# Recent Advancement in Anticancer Compounds from Marine Organisms: Approval, Use and Bioinformatic Approaches to Predict New Targets

**DOI:** 10.3390/md21010024

**Published:** 2022-12-28

**Authors:** Giovanna Santaniello, Angela Nebbioso, Lucia Altucci, Mariarosaria Conte

**Affiliations:** 1Department of Precision Medicine, University of Campania “Luigi Vanvitelli”, Vico L. De Crecchio 7, 80138 Naples, Italy; 2BIOGEM, Institute of Molecular Biology and Genetics, Via Camporeale, 83031 Ariano Irpino, Italy; 3IEOS, Institute for Endocrinology and Experimental Oncology, CNR, Via Pansini 5, 80131 Napoli, Italy

**Keywords:** cancer, marine environment, bioinformatic tools, anticancer compounds, bioactive molecules, molecular mechanism

## Abstract

In recent years, the study of anticancer bioactive compounds from marine sources has received wide interest. Contextually, world regulatory authorities have approved several marine molecules, and new synthetic derivatives have also been synthesized and structurally improved for the treatment of numerous forms of cancer. However, the administration of drugs in cancer patients requires careful evaluation since their interaction with individual biological macromolecules, such as proteins or nucleic acids, determines variable downstream effects. This is reflected in a constant search for personalized therapies that lay the foundations of modern medicine. The new knowledge acquired on cancer mechanisms has certainly allowed advancements in tumor prevention, but unfortunately, due to the huge complexity and heterogeneity of cancer, we are still looking for a definitive therapy and clinical approaches. In this review, we discuss the significance of recently approved molecules originating from the marine environment, starting from their organism of origin to their structure and mechanism of action. Subsequently, these bio-compounds are used as models to illustrate possible bioinformatics approaches for the search of new targets that are useful for improving the knowledge on anticancer therapies.

## 1. Introduction

In recent years, cancer hallmarks have been constantly updated based on tumors’ increasingly complex characteristics. Among these, we can include an emerging hallmark, non-mutational epigenetic reprogramming, which represents another independent mode of genome re-engineering. [1]. Cancer is a set of diseases that dynamically become more and more complex, making the search for effective therapies crucial [2,3]. Lately, several drugs used for the treatment of different types of tumors have been studied, approved and re-proposed alone or in co-treatment. Many of these drugs are of marine origin and are extremely effective for the therapeutic measures of other diseases. There are many reasons that justify the strong interest for molecules of marine origin. Firstly, the sea is a huge source of biodiversity, and thanks to the great variability of conditions present in this environment, its inhabitants have developed many living adaptations. The numerous environmental factors, such as temperature, pressure, pH, light exposure, salinity, gas presence, etc., influence the life of marine organisms also in terms of the complexity of produced molecules [4,5]. Among the strategies to survive these kinds of ecological pressures, there is also the production of secondary metabolites, used as signaling molecules as a form of communication, of defense/offense or for competition [6]. Therefore, the concept of biodiversity offers a major variability regarding molecular interactions and biochemical reactions, and thus, the production of new bioactive molecules that can be potential anticancer drugs [7,8]. This is not enough: it is also crucial to maintain an eco-sustainable marine environment that allows the preservation of these characteristics also in socio-economic terms.

The known marine species, according to the World Register of Marine Species (WoRMS), are about 240,000, and certainly, there are many more, including various kinds of organisms: Animalia, Bacteria, Plantae, Protozoa, Archea, Fungi, etc., but the lack of information does not allow us to trace their real number. From 2010 to date [9], several marine drugs have been approved by the Food and Drug Administration (FDA) and other world regulatory authorities for the treatment of various types of cancer derived from different marine organisms such as: sponges (panobinostat and eribulin mesylate), tunicates (plitidepsin, lurbinectedin and trabectedin) and the mollusk-associated cyanobacteria (brentuximab vedotin; polatuzumab vedotin; enfortumab vedotin; belantamab mafodotin; disitamab vedotin; and tisotumab vedotin) (Figure 1). Depending on their molecular targets and structural features, these compounds share different anticancer activities. Here, we describe the features of these compounds starting from the source and the chemical structure to the different aspects of anticancer activity, also highlighted by the several studies that describe their effectiveness. Starting from these premises, we also investigate potential biological activities related to approved marine molecules by predictive bioinformatic criteria within anticancer therapies. 

## 2. The Evolution of Phenotypic Plasticity in the Marine Environment

Phenotypic plasticity is a phenomenon highly enhanced by environmentally induced alteration [10]. 

Some modifications induced by the environment can influence gene expression and can be also transmitted from a generation to another by a memory mechanism [11,12].

This environmental information makes it possible to respond more quickly to habitat disturbances and represents an adaptative strategy. In particular, organisms that inhabit ecological niches subjected to greater variability are endowed with a greater phenotypic plasticity response both in morphological and physiological terms [13]. There is no direct connection between environmental input and phenotypic output, so phenotypic plasticity cannot always be considered a measure of the reaction norm of an organism. Sessile marine organisms, such as sponges, corals or algae, adapt in the best way to phenotypic plasticity compared to motile organisms through the generation of development variants capable of improving “fitness” generated by the so-called stressors. The study and understanding of all these molecular mechanisms are the basis of new advanced biotechnologies as well as omics and bioinformatics studies. A deeper knowledge of these approaches is at the basis of drug discovery, whose success is closely related to the prediction of new targets and targeted bioassays characterization, as well as high-throughput chemistry.

## 3. Approved Anticancer Compounds from Marine Sponges

Marine sponges belong to the Porifera phylum, and the known species are more than 9000, which are divided into four classes: Calcarea, Demospongiae, Hexactinellida and Homoscleromorpha [14]. To date, more than 5300 natural compounds have been isolated from sponges or their associated bacteria, and this number is constantly updating [15]. Like other sessile marine species, these organisms have developed biological adaptations to live in this habitat and survive predation or fouling by surrounding organisms. The production of secondary metabolites is an important strategy to react to all these conditions, and many of these bioactive compounds can be exploited for human needs [16]. 

From the beginning of the age of marine biodiscovery in the 1960s, some bioactive compounds have been found and approved from marine sources. Among them, the oldest have been isolated from the sponge *Tethya crypta*: the Cytarabine ARA-C discovered in 1959 by Walwick at the University of California used for leukemia and active on DNA polymerase, and the Vidarabine ARA-A is instead an antiviral, which targets viral DNA polymerase used for the first time for the treatment of *Herpes simplex* infection [17].

In the following subsections, we will describe the two most recent approved anticancer drugs related to marine sponges (Table 1 and Appendix A). These compounds are both synthetic analogues–derivatives, which have been developed from natural lead molecules found in these organisms. The possibility to generate other synthetic molecules harboring better pharmacological activity is of extreme importance because these sessile organisms are subjected to continuous biological variability due to abiotic and biotic environmental changes [18].

### 3.1. Eribulin Mesylate

Eribulin mesylate (E7389, Halaven^®^, Eisai Inc., under license from Eisai R&D Management Co., Ltd. © 2022 Eisai Inc. HALA-US3824; Cambridge, MA, USA, us.eisai.com, accessed on 23 October 2022) is the simplified synthetic analog of halichondrin B, a molecule isolated for the first time from the marine sponge *Halichondria okadai* belonging to the Demospongiae class and known for the production of okadaic acid [19]. The cytotoxic activity of halichondrin B was firstly detected in murine models of solid tumors and leukemia, but the low yields obtained from the sponges limited its use [20]. Thanks to the production of synthetic halichondrin B, it was possible to obtain many analogs, such as eribulin mesylate, that are formed by the typical right-hand lactone ring of macrolides, but with the loss of the side chain replaced by a primary amine (Figure 1a). This analog has an optimal activity harboring the original cytotoxic effect of halichondrin B. Eribulin is currently used in clinical approaches for metastatic breast cancer (MBC). In 2010, it was approved in the USA by the FDA as Halaven^®^, and in 2011, the European Medical Agency (EMA) approved its treatment for patients with locally or advanced MBC after two prior anthracycline- and taxane-based regimens [21]. 

This approval is based on a phase III clinical trial (NCT02753595), named EMBRACE, which showed an overall survival advantage of 55% in patients treated with this molecule compared to those that received standard therapy. In 2016, eribulin was also approved by the FDA for the treatment of metastatic liposarcoma and leiomyosarcoma in patients who received a prior anthracycline-containing regimen [22]. A cohort study called ESEMPiO verified the efficacy and safety of eribulin by collecting data from 39 participating centers in Italy including more than five hundred patients that received at least one eribuline treatment. This study confirmed the outcomes observed in the clinical trials, maintaining expected clinical activity and respecting a tolerable safety profile [23]. Eribulin elicits anticancer activity via mitotic and non-mitotic mechanisms of action. It prevents the formation of mitotic spindle-blocking cells in the G2-M phase by inducing apoptosis, but it interferes also with tubulin polymerization suppressing the growth phase of microtubules [24]. Its microtubule-depolymerizing mechanism differs from other tubulin-targeting drugs for several reasons. First of all, it does not affect the shortening phase of tubulin like other vinca alkaloids, but it acts on the growth phase and also binds a higher affinity site compared to taxanes [25]. The resulting studies also showed a different response in the tissues treated with eribulin compared to classical microtubulin-targeting agents; in particular, different effects on peripheral nerves, angiogenesis, vascular remodeling and epithelial-to-mesenchymal transition were detected, affecting differently the tumor microenvironment [26].

### 3.2. Panobinostat

Panobinostat (LBH-589, Farydak^®^, Novartis Pharmaceuticals Corporation East Hanover, New Jersey 07936) is a synthetic analog of Psammaplyn A, which was discovered in 1987 from a Tonga marine Demospongia firstly named *Psammaplin aplysilla* and then revised with the current name *Pseudoceratina purpurea* [27]. Subsequently, it was also found in other species and was soon identified as a promising molecule sharing a great variety of biological activities ranging from antibacterial, antiviral, insecticidal to anticancer activities (Appendix A) [28,29,30,31,32,33]. Psammaplyn A is formed by two symmetrical bromotyrosine structures bonded by a disulfide bridge (Figure 1b). After its discovery, many derivatives were found with different substituent groups, such as psammaplyn B-D and biprasin [34] and psammaplyn E–J [35,36]. The synthetic production of these compounds intensified the research on their potentialities and applications while also considering that, generally, marine organisms cannot be harvested in a massive way to extract compounds.

Panobinostat shares cytotoxicity towards different cancer cell lines such as triple-negative breast cancer, endometrial and prostate cancers [37,38,39], as well as on multiple myeloma (MM) cells [40]. The wide anticancer activity of this drug is due to the effects on many key enzymes involved in different important biological mechanisms in cancer cells: DNA replication and transcription, the regulation of apoptosis, invasion and differentiation. Panobinostat plays an important role in epigenetic regulation involving histone deacetylase (HDAC) and DNA methyltransferase (DNMT) activities [35,39]. It acts as a deacetylase inhibitor able to modulate class I, II and IV of HDAC enzymes by increasing histone and non-histone proteins acetylation, thus, affecting the interactions with transcriptional factors and resulting in the alteration of their functions and the expression of specific genes. The effect of panobinostat in cancer cells has been attested through the acetylation of α-tubulin or HSP90; both events are mediated by the inhibition of HDAC6, influencing the tubulin dynamics and cell motility, as well as affecting the degradation of pro-growth proteins [41,42]. These cumulative effects lead to the inhibition of cellular proliferation and cell-cycle arrest or apoptosis in malignant cells [43]. Another important action of panobinostat in cancer cells is the inhibition of aminopeptidase N, which is involved in different processes, such as proliferation, angiogenesis and tumor invasion [31,44]. Following the PANORAMA-1 phase III trial (NCT01023308), the FDA approved panobinostat in 2015 for patients with relapsed or refractory MM in combination with bortezomib and dexamethasone and after two previous therapeutical regimens. The clinical trial showed an increase in the survival rates in patients treated with panobinostat, corresponding to 5.5 months more, in terms of time, if compared to the untreated group [45] (Table 1 and Appendix A).

**Table 1 marinedrugs-21-00024-t001:** Drugs approved as anticancer compounds obtained from marine sponges: details about sponge species, kind of compound and targets in cancer cells.

Compound Name	Chemical Class	Marine Source	Species	Mechanism of Action	References
Eribulin mesylate	Macrolide	Sponge	*Halichondria okadai*	Interfering tubulin polymerization	[21,22,24,26,46,47,48]
Panobinostat	Hydroxamic acid derivative	Sponge	*Pseudoceratina purpurea*	HDAC inhibitor	[40,43,45,49,50,51]
				Aminopeptidase-N inhibitor	[44]
				DNA Methyltransferase inhibitor	[35]

## 4. Approved Anticancer Compounds from Marine Tunicates

Urochordata, commonly named Tunicates, are a sub-phylum of Chordata divided into five classes, among which the Ascidiacea is represented by 3000 known species, and according to taxonomists, there are more than 1500 species not yet classified [52]. These organisms populate all marine habitats from the low to the deep sea and present a great tolerance to different climate conditions. The huge biodiversity of organisms is connected to a peculiar capability to adapt to different environmental variations and to produce bioactive compounds; to date, there are more than 1200 distinct products characterized in Ascidians [53].

In the following subsections, we will describe the latest three approved drugs from two species belonging to Ascidiacea class (Table 2 and Appendix A). These organisms are characterized by a very small genome and a short life cycle, distinctions which make them very attractive for developmental biology. For these reasons, there is a strong effort to characterize new species and new molecules to understand their pharmacological potential.

### 4.1. Plitidepsin

Plitidepsin (Aplidin^®^, PharmaMar, S.A., Colmenar Viejo, Spain) is a cyclic depsipeptide, isolated for the first time in 1991 from the Mediterranean tunicate *Aplidium albicans*, and belongs to didemnins, a group of molecules studied for their cytotoxic effects (Figure 1c) [54]. Another member of this family is the didemnin B, previously found in the Caribbean *Trididemnum solidum*, but due to its neuromuscular toxicity, studies were stopped, preferring the well-tolerated and more active plitidepsin. The difficulty to harvest and cultivate the organisms slowed down the studies, even if the anti-tumor activity was really interesting [55,56]. Nowadays, thanks to a deeper knowledge of aquaculture systems and cultivation methods and a fine use of the multi-step total synthesis processes, it is easy to produce this depsipeptide, preserving the marine ecosystem. 

The activity of plitidepsin is related to the interaction with the eucaryotic Elongation Factor 1 Alpha 2 protein (eEF1A2), which induces early oxidative stress, causing the rapid activation of c-Jun N-terminal kinase (JNK) and p38/MAPK, leading, finally, to apoptosis [57]. Plitidepsin has been studied for many kinds of cancer, and a promising activity has been detected in the therapeutical approach for MM due to the overexpression of eEF1A2 in affected B cells [58,59]. Plitidepsin was approved in 2018 for refractory MM by the Australian Therapeutic Goods Administration. The phase III (NCT01102426) clinical study, ADMYRE, applied this therapy after the combined use of dexamethasone and at least three prior regimens [60]. The use of this molecule is not extended to other countries because the benefits of the therapy have been considered limited for its side effects, but in 2021, the European General Court annulled the EMA’s previous refusal for market authorization and activated another ongoing judgment of the therapy [61]. Some evidence and trials have analyzed the potential use of plitidepsin as a candidate for the treatment of COVID-19 in specific patients, but this approach needs further investigation [62,63]. Anyway, a case report study recently indicated plitidepsin as a successful treatment for prolonged viral SARS-CoV-2 replication in a patient previously affected by chronic lymphocytic leukemia (CLL). The mechanism of action involved the inhibition of elongation factor 1α (eEF1A) by plitidepsin, which prevents the expression of SARS-CoV-2 nucleocapsid (N) protein [64].

### 4.2. Trabectedin 

Trabectedin (Yondelis®, PharmaMar, S.A., Colmenar Viejo, Spain) was firstly isolated after the NCI screening program of marine natural products in the 1960s by the Caribbean tunicate, *Ecteinascidia turbinata*, but later, it was assessed that its production was due to the associated microorganism *Candidatus Endoecteinascidia frumentensis* [65]. This molecule is an alkaloid, structurally complex and rarely found as a product of marine organisms. Its structure is composed of two rings of tetrahydroisoquinoline (THIQ) fused to another through a lactone bridge, and the full elucidation of this complex structure (Figure 1d) was defined with strong efforts, taking 20 years of studies since its discovery [25,66]. The cytotoxic activity of trabectedin towards cancer cells is given by its interaction with the DNA. The huge molecule alkylates the DNA in the minor groove interfering with proteins and transcription factors and also with molecules involved in DNA repairing processes; as a consequence, there is a perturbation of the cell cycle and induction of apoptosis [67]. An in vitro study analyzed the effect of trabectedin on cancer cell lines by studying the block of RNA polymerase II during elongation. Following transcription, RNA Pol II collides with the trabectedin, reducing the possibility of its movement along the strand. This block causes the ubiquitination of the RNA transcript and induces its degradation via proteasome [68,69].

Trabectedin also induces structural DNA modifications that can affect the recognition of transcriptional factors to specific GC sequences; an example is the inhibition of the binding of TATA-Box Binding Protein (TBP), E2 factor (E2F) and nuclear transcription factor Y (NF-Y) causing the inhibition of the expression of the multidrug exclusion pump, MDR1, associated with chemoresistance [70,71,72].

Another clinical relevance in patients affected by myxoid liposarcomas is the ability of trabectedin to displace the oncogenic transcription factor called FUS-CHOP from its target sequence. This interaction leads to the derepression of the adipocytic differentiation process, revealing an important advantage for the therapy of this kind of cancer [73].

Trabectedin had its first regulatory approval in 2007 from EMA as therapy for patients affected by advanced soft tissue sarcoma after failure of anthracyclines or ifosfamide regimens (NCT01299506). The FDA approved trabectedin therapy in 2015 for liposarcoma and leiomyosarcoma after prior treatment with anthracycline (NCT01343277) [74]. This drug is very effective for liposarcoma treatment, and studies showed a strong reduction in the risk of cancer progression. 

Another way for trabectedin to affect cancer progression is related to the reduction in tumor-associated macrophages (TAM). The strong presence of macrophages in solid tumors, as in breast cancer, enhances tumor growth and the progression of metastasis. Trabectedin reduces the presence of macrophages with an indirect anticancer activity, affecting the tumor microenvironment [75,76].

Some studies also revealed that the use of trabectedin at low concentrations inhibits the production of pro-inflammatory mediators, such as VEGF, IL6, etc., highlighting its role in cytokines and chemokines modulation [75,77].

### 4.3. Lurbinectedin

Lurbinectedin (Zepzelca^TM^, PharmaMar, S.A., Colmenar Viejo, Spain) is another alkaloid that originates from *Ecteinascidia turbinata*. Its structure differs in the composition of the C ring of THIQ, which is replaced by a tetrahydro-β-carboline (Figure 1d). This compound exhibits tumor-inhibiting activities toward metastatic small-cell lung cancer and, in 2020, received FDA approval for patients who received platinum-based chemotherapy (NCT02566993) [78].

The mechanism of action is similar to trabectedin: the drug binds to DNA in the minor groove along GC-rich sequences, interfering with transcription and with repair processes. Despite this, there is the formation of adducts that cause a cascade of events that affects the activity of DNA-binding proteins, transcription factors and repair pathways, leading to eventual double-strand breaks and cell death [79]. 

Lurbinectedin can also inhibit the binding of transcription factors to DNA. EWS-FL1 is an oncogenic transcription factor expressed in pediatric Edwing sarcoma, and it also drives other different kinds of tumors. Xenograft models showed that lurbinectedin can functionally inactivate EWS-FL1, modifying its distribution inside the nucleus and its activity [80]. Lurbinectedin and trabectedin are two derivatives of the same molecule, and their action on cells is very similar, they have two ways to operate their anticancer activity. The first one is the alkylation of DNA characterized by the physical interaction between DNA sequences and the drug, an event strictly connected to the occurrence of DNA damage and strand breaks leading to cell death. A second mechanism involves the interference between DNA interaction and transcription factors by selectively affecting tumor development [81]. Similar to trabectedin, lurbinectedin can decrease tumor-associated macrophages by altering the tumor microenvironment [67].

**Table 2 marinedrugs-21-00024-t002:** Drugs approved as anticancer compounds obtained from marine tunicates: details about species, kind of compound and targets in cancer cells.

Compound Name	Chemical Class	Marine Source	Species	Mechanism of Action	References
Lurbinectedin	Alkaloid	Tunicate/bacteria	*Ecteinascidia turbinata/* *Candidatus* *Endoecteinascidia frumentensis*	Alkylation of DNA	[79]
				Inhibition of the transcriptional factors binding to DNA	[80,81]
Trabectedin	Alkaloid	Tunicate/bacteria	*Ecteinascidia turbinata/* *Candidatus* *Endoecteinascidia frumentensis*	Block RNA Pol-II	[68,69]
				Inhibition of the transcriptional factors binding to DNA	[70,71,72]
				Reduction in TAM	[75,76]
Plitidepsin	Peptide	Tunicate	*Aplidium albicans*	Activation of eEF1A2	[55,82,83,84]

## 5. Approved Anticancer Compounds from Mollusks/Cyanobacteria Association

In the last ten years, many antibody–drug conjugates (ADCs) have been approved, starting from molecules isolated from the symbiosis of cyanobacteria and mollusks (Table 3 and Appendix A).

The number of Molluska marine species is currently estimated over 50000, while cyanobacteria cover over 1600 species and, following some prediction studies, are estimated to be more than 6000 [85,86]. Cyanobacteria embrace lots of symbiotic associations with different marine organisms such as Porifera, Cnidaria, Molluska, Macroalgae, etc., and these symbioses can produce secondary metabolites and also toxins showing bioactivities that can be useful for various treatments, including cancer [87]. 

The analysis of the symbiotic association between these organisms is really interesting, considering also that knowledge about cyanobacteria biosynthetic potentialities is limited, and many of their genomes have not been yet sequenced. Because the exchange of infochemicals is constantly altered, a deeper understanding of chemical ecology coupled with biotechnological advancements could help to acquire more in-depth knowledge.

Thanks to the latest generation technologies, all this is becoming more possible, but certainly, we are still very far from the possibility of being able to somehow standardize the beneficial effects that arise from the association of these marine organisms

### 5.1. Brentuximab Vedotin

Brentuximab vedotin (SGN-35, Adcetris^®^, Seagen Inc., Bothell, WA, USA) leads to another class of anticancer compounds recently approved deriving from dolastatins, linear pentapeptides isolated less than 40 years ago from the sea hare, *Dolabella auricularia*, a mollusk from the Indian Ocean [88]. After the discovery of the active compound, it was clear that its production was due to the presence of two cyanobacteria present in the mollusk-based diet, namely, *Symploca hydnoides* and *Lyngbya majuscula*, members of the order Oscillatoriales [89,90,91].

Due to the toxic effect exhibited by dolastatins, the same studies for structural modifications were necessary to exploit their cytotoxic potentialities [92]. Auristatins are derivatives of dolastatins, composed of four amino acids with different modifications at the C-terminus. Mono-Methyl auristatin-E (MMAE) and Mono-Methyl auristatin-Phe (MMAF) are the most representative among studied drugs, modified respectively with Norephedrine and phenylalanine at their C-termini [93]. These molecules have been associated with monoclonal antibodies, allowing their use as antibody–drug conjugates (ADCs), to better exploit their activity (Figure 1e). In this way, the cytotoxic effect is highly specific and directed to the cells that expose the correct antigen [94].

The mechanism of action is common for all the ADCs associated with auristatin. The antibody recognizes the antigen present on the surface of the cancerous cell and then is internalized through clathrin-mediated endocytosis. Once inside the cell, the lysosome proteases cleave the linker and release the auristatin, affecting its cytotoxic activity.

The auristatin released in the cell acts as an anti-tubulin agent preventing its polymerization, so cell division is inhibited, and apoptosis is induced [95]. These improvements in the use of auristatin derivatives have been applied to many clinical trials for different kinds of cancer diseases; especially, the use of chemo-labeled antibodies has been approved for several molecules.

Brentuximab vedotin is an example of the first commercially marine-derived ADC used and was approved by the American FDA in 2019 for the treatment of patients affected by Hodgkin’s lymphoma, systemic anaplastic large cell lymphoma or cutaneous and peripheral T cell lymphomas (NCT01421667) [96].

This drug is formed by four conjugated molecules of MMAE, also defined as vedotin, bonded via a protease-cleavable linker to the anti-CD30 antibody, normally expressed in activated B and T cells. Its ligand, CD30, is rarely expressed in normal cells, but mainly expressed in diseased cells of patients [97].

### 5.2. Polatuzumab Vedotin

Another approved ADC is Polatuzumab vedotin (Polivy^TM^, DCDS-4501A, Genentech, Inc., San Francisco, CA, USA). In this case, there are 3.5 MMAE molecules (quantity defined following the DAR—drug-to-antibody ratio) associated with CD79b [98]. This antigen belongs to the B cell receptor complex, and its ligand is highly expressed in lymphoma patients’ B cells; moreover, its activation involves the downstream signaling of BRC [99]. This drug was approved by the FDA in 2019 for the treatment of diffuse large B cell lymphoma in combination with two prior therapies including rituximab and bendamustine [100]. This therapy is highly specific and was approved after a phase Ib/II study (NCT01992653) on ineligible patients for transplantation and with previous failure treatment with standard therapies [101].

### 5.3. Enfortumab Vedotin-Eifv

Enfortumab vedotin-eifv (Padcev^®^, AstellasPharma & Seagen Inc., Bothell, WA, USA) is on the market for the treatment of metastatic or locally advanced urothelial cancer. This therapy was approved by the FDA in 2019 after a phase II trial (NCT03219333) for patients that previously received treatment with a platinum regimen and with inhibitors of programmed cell death (PD-1) and programmed death ligand 1 (PD-L1) [102]. The molecule is structurally formed by 3.8 molecules of MMEA—a linker—and the antibody targeting Nectin-4—a transmembrane protein characteristic of epithelial cancers—and used as specific target for Nectin-4-positive urothelial cancer [103]. The approval of enfortumab arrived on April 2022, also in Europe after USA approval, following a phase III trial (NCT03474107) that confirmed a higher overall survival for treated patients of about 4 months [103].

### 5.4. Disitamab Vedotin

Another approved ADC formed by vedotin is the Disitamab vedotin (Aidixi^TM^, RemeGen, Yantai, China). This molecule is associated with the human epidermal growth factor receptor-2 (HER-2) antibody involved in the regulation of cell duplication, proliferation and apoptosis [98]. Since several cancers are HER-2-positive, the antibody present in this molecule is used as a target for anticancer compounds. In this case, anticancer activity is explicated in two different ways: one is the inhibition of signaling pathways activated by HER-2; the other one is operated by MMAE that acts, as mentioned before, as a microtubulin inhibitor blocking mitosis [104]. Disitamab was approved in China by the Chinese National Medical Products Administration for HER-2-positive locally advanced or metastatic gastric cancer after a phase II study (NCT03556345) in patients who received two prior chemotherapeutic treatments [105].

There is also an ongoing phase II study to evaluate the therapeutical possibility of this molecule for breast cancer, comparing the efficacy with different drug doses against HER-2-positive and HER-2 low expression in cancer patients [106]. Urothelial cancer is another typical HER-2-overexpressing cancer, and its response to disitamab has also been evaluated [107] in a phase II study (NCT04264936) on patients with HER-2 locally advanced or metastatic urothelial cancer refractory to classical therapies, which showed good overall survival and a promising efficacy with a manageable safety profile [108].

### 5.5. Tisotumab Vedotin-tftv

Tisotumab (Tivdak^®^, Seagen Inc., Bothell, WA, USA) is a vedotin ADC associated with a specific antibody for the tissue factor TF-011, also called thromboplastin. This cell-surface glycoprotein is normally involved in blood coagulation but is also expressed in many cancer types, such as cervical, ovarian and bladder cancers [109,110,111,112]. The role of TF-011 in tumor progression is supposed to be related to its procoagulant activity and the protease-activated receptor-2 (PAR-2) signaling pathway [113,114].

The cytotoxic activity of tisotumab is due not only to the effect of vedotin but also to the capacity of the molecule to diffuse into the tumor microenvironment and to kill the dividing cells. In this way, tisotumab can bind FCγRIIIa, present on bystander natural killer cells, and leads to antibody-dependent cellular cytotoxicity [115].

Tisotumab was approved in 2021 by the FDA for the therapy of recurrent and metastatic cervical cancer after the phase II InnovaTV 204 trial (NCT03438396) in patients that received two prior systemic regimens, including platinum-based chemotherapy [116], and the recruitment is now ongoing for the phase III trial, InnovaTV 301 (NCT04697628) [117].

### 5.6. Belantamab Mafodotin-blmf

Belantamab mafodotin-blmf (Blenrep^®^ GlaxoSmithKline, Ireland) is a recently approved drug for the therapy of relapsed or refractory multiple myeloma (RRMM) in patients who received at least four therapies, including an immunomodulatory agent, an anti-CD38 monoclonal antibody and a proteasome inhibitor. This therapy obtained regulatory approval from the FDA and EMA in 2020. The DREAMM-2 phase II trial (NCT03525678) demonstrated a durable response to this therapy in RRMM patients. A phase III study is still ongoing, and it will evaluate about 320 patients with more than two prior lines of therapies, whereafter the results will be available in 2024 [118].

Belantamab mafodotin is composed of four molecules of MMAF as the cytotoxic part. MMAF is also defined as mafodotin due to a phenylalanine aminoacidic residue that gives different permeability characteristics to the compound. 

The selected antibody for this ADC is IgG1k, which is specific for the antigen B cell maturating agent (BCMA). In this case, there are two different mechanisms of action: one is given by the effect of mafodotin, and the other is given by the antibody. This antibody is particularly active because it is afucosylated, and this characteristic enhances the recruitment of immune effector cells because afucosylation creates a higher binding affinity for the receptor, FCγIIIaR, which is also involved in the interaction with the B cell. In this way, the molecule can kill tumor cells by both antibody-dependent cellular cytotoxicity and antibody-dependent cellular phagocytosis [119,120]. 

**Table 3 marinedrugs-21-00024-t003:** Drugs approved as anticancer compounds obtained from mollusk-associated cyanobacteria: details about species, kind of compound with associated antibody and targets in cancer cells.

Compound Name	Chemical Class	Marine Source	Species	Mechanism of Action	References
Brentuximab vedotin	ADC (MMAE + CD30Ab)	Mollusk/ Cyanobacteria	*Dolabella auricolaria/* *Symploca hynoides, Lyngbya majuscula*	Microtubulin targeting agent via CD 30	[95,96,97]
Polatuzumab vedotin	ADC (MMAE + CD-79bAb)	Mollusk/ Cyanobacteria	*Dolabella auricolaria/ Symploca hynoides, Lyngbya majuscula*	Microtubulin targeting agent via CD79	[100,101,121]
Enfortumab vedotin	ADC (MMAE + Nectin-4 Ab)	Mollusk/ Cyanobacteria	*Dolabella auricolaria/ Symploca hynoides, Lyngbya majuscula*	Microtubulin targeting agent via Nectin4	[103,122]
Disitamab Vedotin	MMAE + HER-2 Ab	Mollusk/ Cyanobacteria	*Dolabella auricolaria/ Symploca hynoides, Lyngbya majuscula*	Microtubulin targeting agent via HER-2	[105,107,108,123]
Tisotumab Vedotin	MMAE + TF-011 Ab	Mollusk/ Cyanobacteria	*Dolabella* *auricolaria/ Symploca hynoides, Lyngbya majuscula*	Microtubulin targeting agent via TF-011	[113,114,115,116]
Belantamab mafodotin	MMAF + CD38Ab	Mollusk/ Cyanobacteria	*Dolabella auricolaria/ Symploca hynoides, Lyngbya majuscula*	Microtubulin targeting agent via CD38	[118,119,120]

## 6. Prediction Bioinformatics Tools in Marine-Derived Compounds

Here, we evaluate the therapeutic significance of the molecules derived from marine species described in the previous sections to subsequently identify further biological activities that could be predictive for the study of the molecular mechanisms involved in cancer progression. 

The use of several tools and software connected to databases is an advantageous approach that avoids different issues, such as the expensive costs of screening or the time spent to exploit the in vivo capabilities of compounds [124]. In silico methods are a valid opportunity to analyze the possible activities of chemicals for a new vision of drug discovery, which allows identifying the cytotoxicity of molecules or checking the activation of specific metabolic pathways. It could represent a possibility to save time in research study, giving an early result based on a prediction. This is a not dismissible point, considering the need for new drugs for several kinds of diseases.

One of the tools we used for these studies is PASS (Prediction of Activity Spectra for Substances (http://www.way2drug.com/passonline/, accessed on 10 October 2022, a software based on the Bayesian method that, starting from the structure of the molecule, can give accurate and robust predictions about a possible bioactivity [125]. We used the SMILE format (simplified molecular-input line-entry system), which expresses molecular formulas and reactions in line notation, adopting letters for the atoms, symbols (-, = or #) for the bonds, enclosed parentheses for the branches and specific rules for cyclic and disconnected structures. The use of this language is very common because it is compact and printable and takes less space than an equivalent connection table, also in terms of bytes for the use of databases or software. There are different formats used to upload molecules to these databases; one is SMILE, but MOL and Marvin JS files are commonly accepted by these kinds of software, which represent, respectively, the table format and the image of the structure of the studied molecule. These molecule formats are available in commonly used databases, such as PubChem https://pubchem.ncbi.nlm.nih.gov/ (accessed on 10 October 2022, and it is also possible to use online converters to obtain the different formats (such as https://cactus.nci.nih.gov/translate/, accessed on 10 October 2022. The different ways to express the analyzed molecules is listed in Appendix A.

To give an example of the application of these bioinformatic tools, we selected three of the molecules analyzed in the previous section, one for every marine origin group.

This kind of analysis matches data contained in different databases, starting from the information present in the structure of the compound and also the knowledge available in the literature. The first evaluation involved the activity spectrum of every single molecule according to two distinct parameters defined by the software, considering the probability of being active (*Pa*) and the probability of being inactive (*Pi*).

We applied a cut-off of *Pa* > 0.5 to these values as a general parameter for all the analyzed predictions. We also considered the use of another service related to PASS, named CLC-pred (cell line cytotoxicity predictor), to predict the cytotoxic effect of a selected compound in cancer and non-cancer cell lines. This information will drive the choice of future target cancer cell lines to select the bioactivity of the drug-like compound [124].

Another point we evaluated regarded the possibility of obtaining data related to the clinical manifestations observed in patients, including toxic and adverse effects. Once these parameters were identified using the DIGEP tool (prediction of drug-induced changes in gene expression profiles) [126], DIGEP was connected to PASS by accessing an external link and matching with different databases provided distinctive information about the effects of the drug on cell metabolism and about the changes in gene expression. Using different training sets, we identified differentially up-regulated and down-regulated genes following the level of mRNA and expressed proteins after drug treatments. The results were extrapolated matching the data present in the current literature; the mRNA expression profiles were obtained by exploiting the mRNA training set of the tool, composed of 1756 compounds, which allows one to predict the changes in gene expression for 1802 genes. 

The obtained data were further evaluated using a graphical gene set enrichment interactive tool [127], wherein queried genes, using Ensembl code, were analyzed based on gene ontology (GO) biological processes and molecular function, using ShinyGo (http://bioinformatics.sdstate.edu/go/, accessed on 24 October 2022 by selecting the species homo sapiens. In the following subsections, we will make an exemplary analysis of each compound described for the corresponding kinds of organisms using these tools. Figure 2 provides a schematic representation of the bioinformatic tools used and the specific outcomes. 

### 6.1. Sponge-Derived Compound Prediction: The Example of Panobinostat

We added to the PASS database the structural formula of panobinostat by using SMILE and launched the prediction. As expected, the data obtained showed a strong Pa relative to the different classes of HDAC inhibitors, as confirmed by the vast evidence already present in the literature and discussed in the previous dedicated section. We also identified panobinostat-dependent activities as chemosensitizers and activators of the calcium channel but with lower probability (Table 4). Anyway, recent studies have suggested the use of panobinostat as a chemosensitizer in advanced ovarian cancer [128]. No significative data, instead, are reported concerning the activation of the calcium channel by panobinostat.

We also predicted the toxic effects based on the possible clinical manifestations and the different cancer cell lines, which could result from sensitivity to treatment by CLC-pred. Panobinostat shared low *Pa* values for adverse effects such as ulcers or the presence of occult bleeding, whereas considering the possible cancer lines sensitive to panobinostat treatment, we mainly observed a *Pa* value of 0.784 for HCT-116 colon carcinoma cells (Table 5 and Table 6). In particular, from the data obtained through the consulted databases, panobinostat-related possible cytotoxic effects were found in the colon cancer line, HCT-116, as previously indicated and with regard to its combined effect and its safety profile. In particular, the TRAIL-induced cytotoxic effect is enhanced by co-treatment with panobinostat in HCT-116 [129]. 

Considering the predictions made for the interaction with other cancer cell lines, we observed that in the lung cancer cell line, NCI-H1299, panobinostat displayed a *Pa* = 0.543; while in the triple-negative-resistant breast cancer line, the cytotoxic effect was detected with a *Pa* = 0.518. Identified in silico predictions by PASS suggested that, despite the fact that panobinostat was initially identified for the treatment of MM, it may be considered a good candidate for the study and treatment of other diseases.

Using DIGEP as a prediction system we matched the genes involved in the metabolic pathway activated by the drug, considering the data present in the comparative toxicogenomics database (CDT). In Table 7, several up- and down-regulated genes are predicted to be differentially expressed following panobinostat treatment. Notably, in a recent study, the *CTPS1* gene, which shares high *Pa* prediction in our analysis, was associated with a lower survival outcome in plasma cell myeloma patients belonging to the high-risk metabolic cohort [130]. This is very interesting information that can be extrapolated from our prediction since panobinostat could represent a valid candidate for a better understanding of metabolic-based mechanisms linked to this disease.

Starting from the data obtained from the DIGEP predictions, the genes whose expression is influenced by the drugs were submitted to GO analysis. Data showed for ShinyGo analysis are based on the analysis of normal cell lines in human models not treated with the drug. Both up- and down-regulated genes were then analyzed by the ShinyGo tool in order to identify possible predictive GO biological and molecular functions. For both GO processes, we applied a false discovery rate (FDR) cut-off of 0.05. Data were represented by each category name related to fold enrichment −log10 (FDR). As we can see in Figure 3A, the higher bar size is represented by the category of genes that regulate the cytoplasmatic sequestering of transcription factors, suggesting that panobinostat selectively is able to interact with specific transcription factors in the cytoplasm, which might be considered a predictive effect of this molecule in this specific biological compartment. For example, panobinostat is involved in the regulation of transcription factors present in the HOXA cluster, which are under the control of the mixed-lineage leukemia (MLL) complex, whose proteins are fused or mutated through interactions with many cofactors [131]. 

This demonstrates its role also in off-target interactions that can be identified through predictive analyses and provide useful information to better understand the molecular mechanisms underlying these regulations. We also identified that panobinostat-related genes might be involved in specific molecular functions, which could also be predictive for different in-depth studies. For instance, we identified genes involved in diamine N-acetyltransferase, phenanthrene 9,10 monooxygenase, polyamine-binding and methotrexate-binding activities (Figure 3B), which are all related to molecular reaction/interaction networks involved in the metabolism. This could mean that a network-based approach could be useful to correlate deregulated pathways and drug effects for a further redout in individual metabolism. These predictions consist of several cellular activities that could be carried out by panobinostat and could offer useful information for a future approach for a better in-depth study of a selected disease. 

### 6.2. Tunicate-Derived Compound Prediction: The Example of Plitidepsin

The same analysis was conducted on plitidepsin by predicting a high activity as an immunosuppressant (*Pa* = 0.788) and also a glycopeptide-like antibiotic (*Pa* = 0.738), general pump inhibitor (*Pa* = 0.649) and, as we know, an antineoplastic (*Pa* = 0.657) (Table 8). This marine cyclic peptide is a clear example of therapeutic switching, which is becoming an increasingly used strategy for the identification of new uses of an approved drug, despite the fact that, after a long time since its discovery, it has been repurposed for the treatment of COVID-19 to answer a current need not evidenced before. 

Analyzing the predictions of the possible adverse reactions and side effects belonging to plitidepsin, we observed a high *Pa* value related to the category of movement disorders identified as dyskinesia, as well as sleep disturbance, dyspnea and ataxia, with *Pa* values ranging from 0.960 to 0.776 (Table 9).

Concerning the predictions of the cytotoxicity towards cancer cell lines, the tool identified relevant results for the lung carcinoma cell line, A549, and also for the colon adenocarcinoma, HT-29, cancer cell line (Table 10). Plitidepsin case studies on its activity towards A549 and HT-29 are few; for example, an investigation was conducted to evaluate the efficacy of the drug encapsulated in some polymers [132]. This tool also identified possible cytotoxic activities in other cancer cell lines, such as breast (MDA-MB-231) and lung (DMS-114) adenocarcinoma cells. Although studies on plitidepsin cytotoxic activity in MDA-MB-231 cells have already been published, information on the activity of this compound in the identified lung cancer line are not available in the literature. This provides us future important indications of the potential cytotoxic effect of this molecule for the study and treatment of this different cellular system.

From the analysis with the DIGEP tool, we identified many up-regulated and down-regulated genes with a high value of *Pa* (Table 11). These genes are highly depicted in metabolic pathways such as those involved in glucose homeostasis. Remarkably, we identified *TMEM41B*, a transmembrane protein-coding gene involved in the host’s modulation of viral RNA genome replication. This is further evidence of plitidepsin-related antiviral activities since recent studies indicated *TMEM41B* as a pan-flavivirus host factor, which is also crucial for SARS-CoV-2 infection [126].

Both up and down-regulated genes were analyzed by the ShinyGo tool in order to identify possible predictive GO biological and molecular functions related to drug treatment. For both GO processes, we applied less stringent data by a false discovery rate (FDR) cut-off of 0.11. Data were represented by each category name related to fold enrichment −log10 (FDR). As can be seen in Figure 4A, the higher bar size is represented by the genes belonging to cellular response to glucose stimulus, suggesting that plitidepsin might have a role in adaptative cellular responses such as those concerning glucose metabolism. It is also known that plitidepsin can affect cellular metabolism by acting on cellular glutathione homeostasis, converting glutathione in the reduced forms and altering its control level mechanisms [130]. We also identified that molecular functions, such as peptide–proton symporter, phosphotransferase and ganglioside binding activities have been recognized as predictive for plitidepsin treatment. Because ganglioside, an anionic lipids family, is also present in this case, we can appreciate a close correlation between the plitidepsin and SARS-CoV-2 interaction by a mechanism involving the recognition of spike proteins. A great deal of predictive data, concerning the role of plitidepsin in the recently discovered COVID-19 pandemic, clearly suggest the importance of using in silico approaches, which is also underlined by the consistent bioinformatics data that are constantly updated, thereby allowing the consultation of recently added biological activities of our molecule of interest.

### 6.3. Mollusks/Cyanobacteria Association-Derived Compound Prediction: The Example of Belantamab Mafodotin

Here, we discuss the last example, analyzing belantamab mafodotin. From the results of the prediction, we observed high values of *Pa*, 0.823, related to immunostimulant activity but also interesting values with proteasome ATPase inhibitor, 0.774, and, as we expected, antineoplastic activity with a 0.729 *Pa* value (Table 12). As we can observe, this compound also shares its activities in solid tumors and, specifically, in pancreatic cancer, despite its approval for MM treatment. Considering the enormous complexity of cancer, these features could be useful in investigating an off-target therapeutic role of this molecule alone or in combination with other therapies.

The prediction for the toxic and adverse effects gave no results in this compound, probably due to the early discovery of this molecule, which did not permit the collection of enough data.

Considering the predictions for the cytotoxic activity towards cancer cell lines, we observed two results with Pa values of 0.501 and 0.513 for two breast cancer cell lines, namely, MDA-MB-231 and MCF7 (Table 13). Because a wide-spectrum of information relative to belantamab mafodotin is correlated to its biological activity in MM, this prediction could be useful for in vitro studies to understand the potential mechanisms underpinning different phenotype-associated breast cancers. We also identified from the DIGEP prediction up-regulated and down-regulated genes following the results of mRNA levels (Table 14). The genes identified are deeply involved in metabolic processes. Interestingly, among down-regulated genes, we identified *ALDH18A1*, which is an allelic variant of the aldehyde dehydrogenase family. Since previous studies showed that the activity of ALDH1, a member of ALDH superfamily, was increased in MM stem cells [128], the identification of its related variant by predictive analysis could be useful for a better understanding of the role of the mutated forms of related genes involved in not only metabolic dysregulation associated with MM but also with other diseases.

With the ShinyGo tool, we predicted the possible GO biological and molecular functions. We applied an FDR cut-off of 0.05 for both biological processes and molecular functions. Data were represented by each category name related to fold enrichment -log10 (FDR). As we can see in Figure 5A, the higher bar size in the biological process is represented by the category of genes that are involved in regulating the sesquiterpenoid metabolic process and the regulation of the isoprenoid process. On the other hand, one of the most representative molecular functions was related to phosphotransferase activity (Figure 5B). Moreover, since we detected breast systems, such as MDA-MB-231 and MCF7, as possible sensitive cancer cell lines, in vitro study could be deepened to extrapolate additional information on these cancer-associated signatures even if the clinical benefits have not been confirmed as in the case of MM. 

## 7. Conclusions 

Cancer diagnostics and treatments have reached a very high degree of complexity in recent years, also considering the interaction between the different fields of study, the approaches used to gather knowledge and the information available for the search for anticancer drug candidates [129]. Subsequently, drugs from increasingly specific natural and synthetic sources have been developed, making possible the identification of important biomarkers that are able to improve the appropriacy of therapies [111]. Marine pharmacology is a rapidly growing discipline [16] thanks to the integration of biotechnologies based on marine organisms and the preclinical and clinical applications of produced natural compounds, such as primary and secondary metabolites. Currently, there are many drugs on the market with active ingredients of marine origin with various therapeutic activities including anti-cancer ones [25]. As described on “the marine pharmaceuticals website” (https://www.marinepharmacology.org/, accessed on 10 October 2022), many marine compounds are currently in phase 1, 2 and 3 trials, while 17 studies have been approved by global regulatory authorities for the treatment of a wide range of diseases [130]. In the first part of this review, we deeply discussed the current literature about the latest approved anticancer biomolecules of three different kinds of marine organisms. We introduced the marine sources, the molecule structures and the biological potentialities, as well as the optimization of the compound for therapeutical efficacy and approval. On the basis of the previously described information, in the second part of the review, we selected three molecules from each related organism as a model to offer an overview about the possible bioinformatic applications for a better understanding of their biological activity. Thus, we firstly discussed the use of an online software, PASS, to evaluate the general biological potential of the described drugs. Thanks to simultaneous predictions based on the structure of the interrogated compounds, we were able to estimate the effectiveness of molecules also in the pathological contexts different from their current therapeutic use. Once the drug-related biological activities were predicted, we consulted the PASS-connected service, DIGEP-pred, to predict drug-induced gene expression, and finally, we examined GO biological and molecular functions with the ShinyGo tool. As for panobinostat, many predicted HDAC inhibitory activities were confirmed by the used tools, but few studies reported its activity as a chemosensitizer or calcium channel activator. Despite preclinical studies, panobinostat confirmed its biological potentials in MM; this drug was predicted to be active also in colon, lung and breast cancer cell lines as a single agent and also in combination with other therapies for solid tumors [39]. Differentially expressed genes associated with panobinostat were detected, indicating its metabolic role in different diseases. Plitidepsin was another analyzed molecule, and it represents a concrete example of a re-proposed drug used recently for the treatment of COVID-19 unlike other re-proposed drugs, such as lopinavir–ritonavir, hydroxychloroquine, remdesivir and interferon beta 1a, which were supposed to be inefficient [131,132]. Consulting the tools, many broad-spectrum plitidepsin-related biological activities were detected to be in line with previous findings [55]. Interestingly, among differentially expressed genes regulated by plitidepsin, *TMEM41B* was identified as a common target for SARS-CoV-2 infection.

The last molecule interrogated in our tool was belantamab mafodotin, which showed from predictions immunostimulant and proteasome ATPase inhibitor activities. Belantamab, approved for RRMM, seemed to share interesting previsions about cytotoxic effects also towards the MDA-MB-231 and MCF7 breast cancer cell lines, suggesting its potential effect also in solid tumors. From the DIGEP predictions, we detected many differentially regulated genes, and among these, we focused our attention on the down-regulated *ALDH18A1* gene, which could be a potential alternative target for MM [128].

Some of the predicted activities we found are partly in line with the already present knowledge thanks to the increase in deposited big data that can help a better characterization of the clinical–pathological features of treated patients.

The obtained results could be useful for giving new information about already known and discovered compounds but can also give suggestions about other possible pharmacological applications or new targets (e.g., the evaluation of different cancer cell lines that are sensitive to the compound). The predictions of the reported three case studies represent an example of the application of these tools, which can give additional information about already known compounds and also suggest further insights to help following studies on new bioactive compounds of marine origin. 

## Figures and Tables

**Figure 1 marinedrugs-21-00024-f001:**
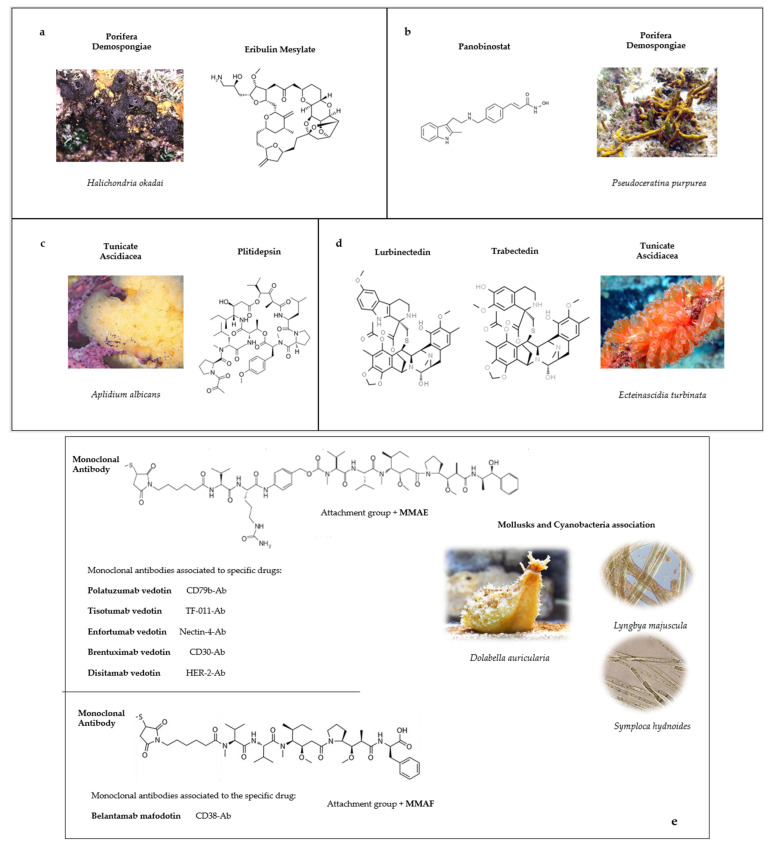
Approved marine drugs for cancer treatment and their organism of origin. (**a**,**b**) Structures and names of compounds derived from marine sponges; (**c**,**d**) structure and names of compounds obtained from tunicates; and (**e**) structures of molecules obtained from the relationship between mollusks and cyanobacteria with specific associated monoclonal antibodies.

**Figure 2 marinedrugs-21-00024-f002:**
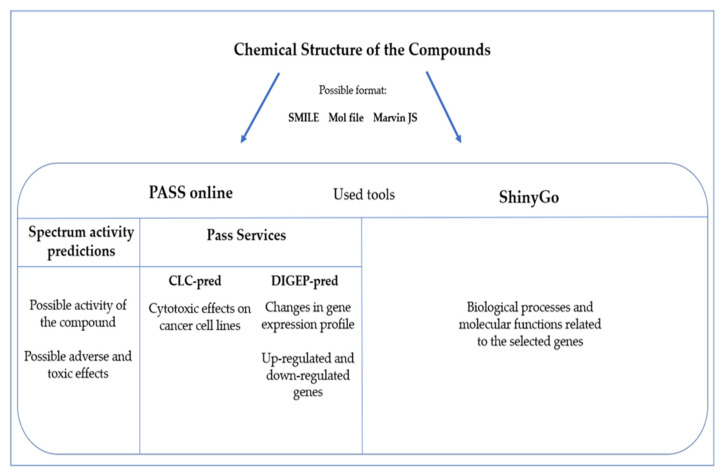
Schematic representation of the pipeline used for the bioinformatic predictions. Selected compounds were analyzed with PASS online and ShinyGo, using three possible formats for the analysis (SMILE, Mol file and Marvin JS). PASS online gives the first prediction about the activity, and then, there are other additional services (CLC-pred and DIGEP-pre are the used ones). ShinyGo is the other used tool for GO predictions.

**Figure 3 marinedrugs-21-00024-f003:**
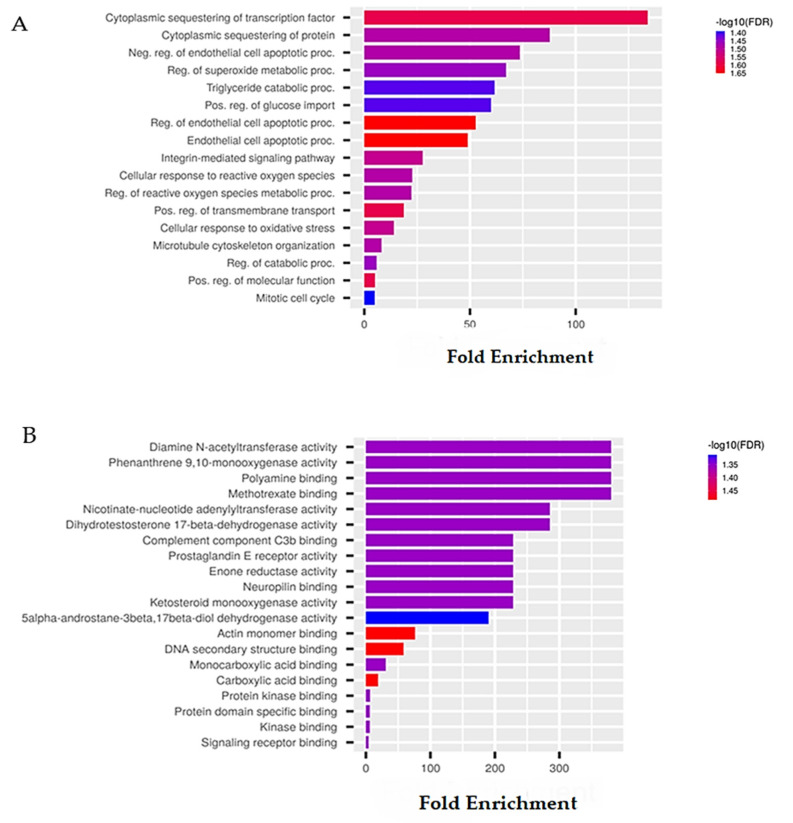
GO predictive analysis of differentially expressed panobinostat-related genes. (**A**) Biological processes on queried panobinostat-dependent genes. Data show each pathway sorted by category related to fold enrichment with -log10 (FDR) and applying an FDR cut-off of 0.05. (**B**) Molecular functions on queried panobinostat-dependent genes applying an FDR cut-off of 0.05.

**Figure 4 marinedrugs-21-00024-f004:**
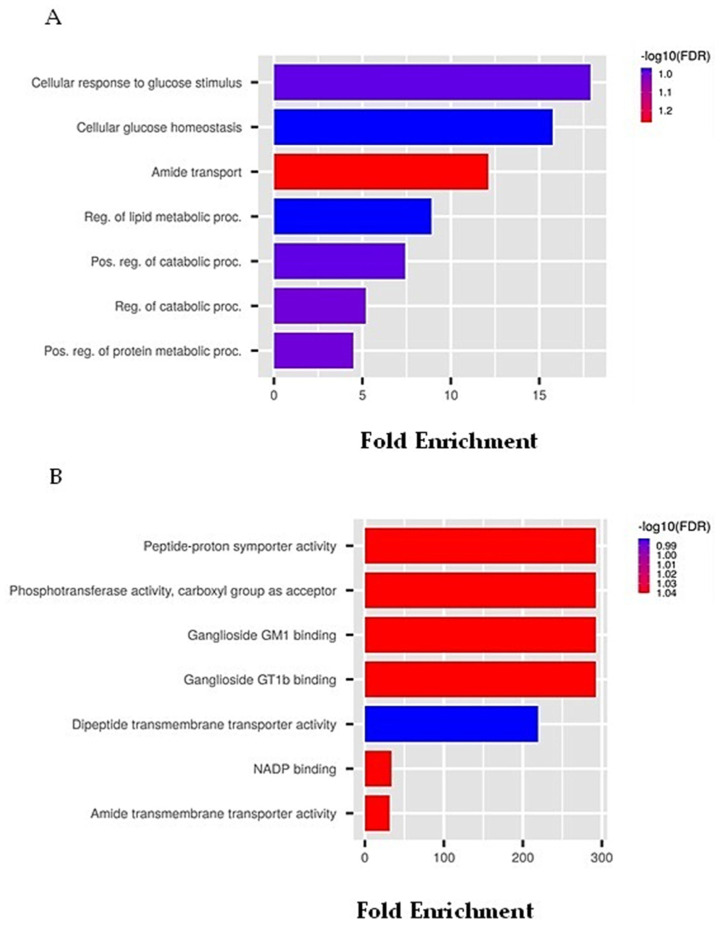
GO predictive analysis on differentially expressed plitidepsin-related genes. (**A**) Biological processes on queried plitidepsin-dependent genes. Data show each pathway sorted by category related to fold enrichment with -log10 (FDR) and applying an FDR cut-off of 0.11. (**B**) Molecular function on queried plitidepsin-dependent genes. Data show each pathway sorted by category related to fold enrichment with -log10 (FDR) and applying an FDR cut-off of 0.11.

**Figure 5 marinedrugs-21-00024-f005:**
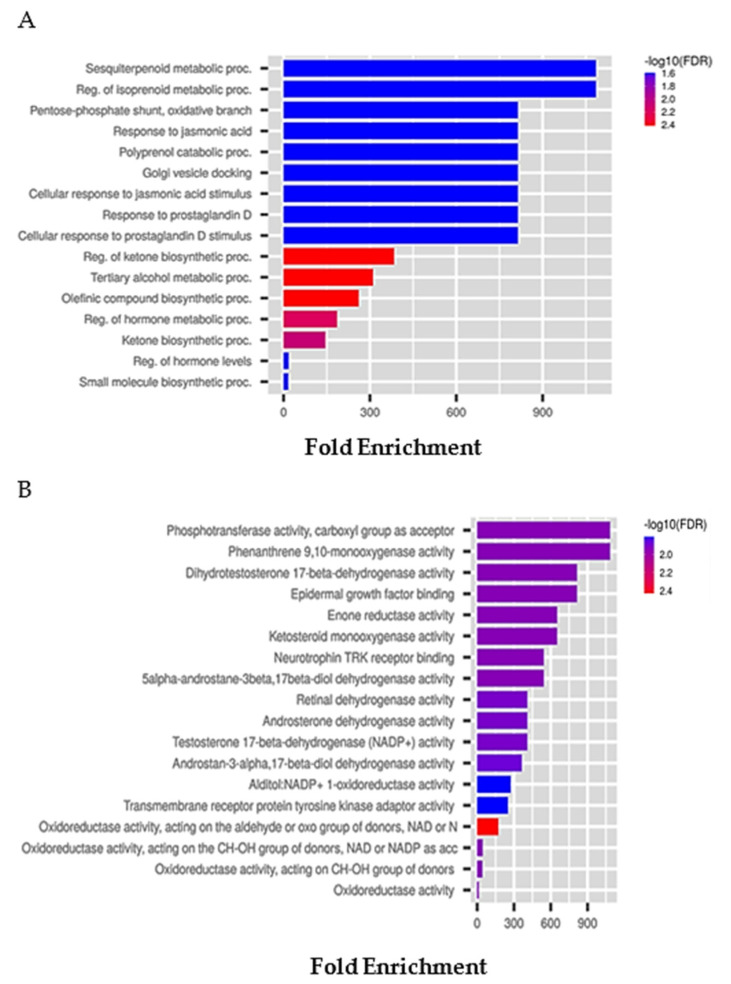
GO predictive analysis of differentially expressed belantamab-related genes. (**A**) Biological processes on queried belantamab-dependent genes. Data show each pathway sorted by category related to fold enrichment with -log10 (FDR) and applying an FDR cut-off of 0.05. (**B**) Molecular function on queried belantamab-dependent genes. Data show each pathway sorted by category related to fold enrichment with -log10 (FDR) and applying an FDR cut-off of 0.05.

**Table 4 marinedrugs-21-00024-t004:** List of the predicted activities recognized by the PASS tool for panobinostat with the indicated probability of being active and inactive from 0 to 1.

Activity	*Pa*	*Pi*
HDAC 1 inhibitor	0.843	0.001
HDAC class I inhibitor	0.842	0.001
HDAC inhibitor	0.792	0.002
HDAC 2 inhibitor	0.760	0.001
HDAC 4 inhibitor	0.706	0.001
HDAC IIa inhibitor	0.700	0.001
HDAC 8 inhibitor	0.530	0.001
Chemosensitizer	0.523	0.016
Calcium channel (voltage-sensitive) activator	0.521	0.064

**Table 5 marinedrugs-21-00024-t005:** Predictions of the possible clinical manifestations observed in patients following panobinostat treatment.

Possible Adverse and Toxic Effects	*Pa*	*Pi*
Ulcer, gastric	0.362	0.057
Occult bleeding Ulcer, peptic	0.421 0.346	0.131 0.088

**Table 6 marinedrugs-21-00024-t006:** Prediction of the possible cancer cell lines sensitive to panobinostat treatment related to cytotoxic activity.

Cell Line Full Name and Code	Tissue	*Pa*	*Pi*
Colon Carcinoma HCT-116	Colon	0.784	0.0070
Non-small cell lung carcinoma NCI-H1299	Lung	0.543	0.004

**Table 7 marinedrugs-21-00024-t007:** DIGEP predictions of panobinostat-regulated genes.

Down-Regulated Genes	*Pa*	*Pi*	Up-Regulated Genes	*Pa*	*Pi*
*CTPS1* *INCENP* *KEAP1* *AKR1C3* *EIF4G2* *RCC2* *ABL1* *PHF11* *TOB1* *DHFR* *FABP4* *MAPK4* *NPM*	0.875 0.863 0.795 0.802 0.700 0.697 0.676 0.683 0.715 0.701 0.664 0.563 0.568	0.001 0.006 0.008 0.022 0.003 0.002 0.002 0.013 0.050 0.048 0.049 0.114 0.137	*TACC1* *ITGAM* *H1F0* *HPGD* *FGF21* *TMSB4X* *OCLN* *SAT1*	0.833 0.819 0.759 0.748 0.686 0.659 0.573 0.592	0.004 0.021 0.005 0.032 0.01 0.034 0.039 0.129

**Table 8 marinedrugs-21-00024-t008:** List of the predicted activities recognized by the PASS tool for plitidepsin with the indicated probability of being active and inactive from 0 to 1.

Activity	*Pa*	*Pi*
Immunosuppressant	0.788	0.006
Antibiotic glycopeptide-like	0.738	0.003
General pump inhibitor	0.649	0.014
Antineoplastic	0.657	0.034
CYP2H substrate	0.650	0.050
Antifungal	0.585	0.020
Antineoplastic (colorectal cancer)	0.548	0.010
Antineoplastic (colon cancer)	0.541	0.010
Xenobiotic-transporting ATPase inhibitor	0.524	0.009
Antibacterial	0.453	0.021

**Table 9 marinedrugs-21-00024-t009:** Predictions of the possible clinical manifestations observed in patients following plitidepsin treatment.

Possible Adverse and Toxic Effects	*Pa*	*Pi*
Dyskinesia Sleep disturbance Dyspnea Ataxia	0.960 0.898 0.892 0.776	0.004 0.012 0.006 0.013

**Table 10 marinedrugs-21-00024-t010:** Prediction of the possible cancer cell lines sensitive to plitidepsin treatment for cytotoxic activity.

Cell Line Full Name and Code	Tissue	*Pa*	*Pi*
Lung carcinoma A549 Colon adenocarcinoma HT-29 Breast adenocarcinoma MDA-MB-231 Lung carcinoma DMS-114	Lung Colon Breast Lung	0.808 0.801 0.554 0.501	0.011 0.005 0.020 0.038

**Table 11 marinedrugs-21-00024-t011:** DIGEP prediction of plitidepsin-regulated genes.

Down-Regulated Genes	*Pa*	*Pi*	Up-Regulated Genes	*Pa*	*Pi*
*NSF* *ALDH18A1* *H6PD* *SLC15A1* *MYBL1* *BACE1* *SLC14A1* *TOB1* *VTN* *BARD1* *FKBP5* *CTPS1* *HSPB11* *TAGLN*	0.907 0.879 0.782 0.765 0.739 0.697 0.675 0.682 0.605 0.636 0.599 0.566 0.501 0.516	0.012 0.015 0.019 0.036 0.040 0.016 0.053 0.063 0.036 0.120 0.086 0.090 0.053 0.088	*TMEM41B* *C10ORF118* *FAM49A* *PLXNA2* *HMGCR* *PSAP* *TXNDC9* *PLK3* *FGF21* *C8ORF4* *WIPI1* *GPRC5A* *NUCB2*	0.823 0.826 0.756 0.751 0.637 0.635 0.542 0.595 0.517 0.552 0.530 0.507 0.523	0.009 0.012 0.041 0.047 0.035 0.055 0.025 0.092 0.034 0.083 0.106 0.087 0.105

**Table 12 marinedrugs-21-00024-t012:** List of the predicted activities by PASS tool following belantamab mafodotin treatment with the probability of being active and inactive from 0 to 1.

Activity	*Pa*	*Pi*
Immunostimulant	0.823	0.008
Proteasome ATPase inhibitor	0.774	0.007
Antineoplastic (non-Hodgkin’s lymphoma)	0.729	0.003
Muramoyltetrapeptide carboxypeptidase inhibitor	0.636	0.021
Antineoplastic (solid tumors)	0.537	0.010
Peptide agonist	0.537	0.039
Neuropeptide Y4 antagonist	0.461	0.027
Antineoplastic (pancreatic cancer)	0.437	0.010
CYP2H substrate	0.502	0.122
Fibroblast growth factor agonist	0.419	0.056

**Table 13 marinedrugs-21-00024-t013:** Prediction of the possible cancer cell lines sensitive to belantamab mafodotin treatment for cytotoxic activity.

Cell Line Full Name and Code	Tissue	*Pa*	*Pi*
Breast adenocarcinoma MDA-MB-231 Breast carcinoma MCF7	Breast Breast	0.501 0.513	0.028 0.049

**Table 14 marinedrugs-21-00024-t014:** DIGEP prediction of belantamab mafotin-regulated genes.

Down-Regulated Genes	*Pa*	*Pi*	Up-Regulated Genes	*Pa*	*Pi*
*ALDH18A1* *SHC1* *NSF* *H6PD* *AKR1C3*	0.698 0.550 0.546 0.524 0.511	0.098 0.040 0.117 0.127 0.211	*C10ORF118* *TMEM41B*	0.631 0.606	0.063 0.051

## Data Availability

Not applicable.

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
