# Peer review of "Recent Advancement in Anticancer Compounds from Marine Organisms: Approval, Use and Bioinformatic Approaches to Predict New Targets"

_marinedrugs, 2022, doi:10.3390/md21010024_

Reviewer 1 Report
General overview
In this review, the authors discuss the potential of the marine environment for cancer drug discovery. In concrete, they review the significance of recently approved molecules originating from marine organisms, focusing on those recently approved originated from three marine sources: sponges, tunicates, and cyanobacteria associated with mollusks. The authors go on describing the organism of origin, their structure and known mechanism of action for each mentioned compound.
This information was then subsequently used, as model to illustrate some possible bioinformatics approaches for searching for new targets useful for improving anticancer therapies. The authors selected 3 case studies to be tested by the bioinformatics approach they designed, with the aim to, not only validate the virtual screening approach taken (expecting similar results than those known today for the used marine compounds), but also, to shed some light into potential new uses of these tools while providing further insights into new research avenues and/or applications for the tested molecules.
To perform this analysis the authors started from the structural characteristics of marine drugs approved by the FDA, to test them against large sets of big data to broaden our knowledge of further mechanisms involved in tumor progression. Authors have identified further targets, and uncover putative novel molecular targets that could be usefull to understand the mechanisms related to tumor progression and predictive of novel therapeutic avenues using these marine derived molecules. Moreover, the data presented can help others to test and predict possible new biological processes and molecular functions involved in those tumors and MoA that might be involved in following treatments with these drugs. Therefore, this review provides a novel dataset for others to develop while gathering the most recent data on marine drugs for cancer treatment.
The review is pertinent and updated and the information provided is mostly accurate.
The citations are relevant and updated.
General comments for improvement:
The abstract and initial introduction focus a significant part in the epigenetics of cancer and environmental epigenetics aspects of marine survival strategies of its inhabitants and its relation to their metabolite and bioactive’s production, but these epigenetic aspects seem to then disappear through the remaining text, so it is unclear why this is mentioned here or which relation it has to the specific topic of this review or the drugs related.
On the other hand, and this being a marine journal and a marine focused paper, this reviewer feels that more attention and detail could be given to the marine component of this review, still focusing on marine cancer drug discovery, with more figures and data of each specific organism(s) involved, their marine preferential locations, type of habitats, etc. Bringing more data on other marine cancer molecules, even if older would increase the depth of this review.
In the bioinformatic analysis section, more context on the described results and potential implications are missing and the authors should relate the described results and outputs with additional literature and potentially discuss future research avenues and implications. This would enrich the discussion part of this review.
Despite these general comments, the review is well designed and addresses well the selected topics, is written in a fluent manner and provides novel insigths into the thematic as stated above.
Specific comments by Line number:
123 – Incorrect spelling of the Sponge name and missing the Italic that shall be used for correct scientific spelling of any species name. It should be replaced by Halichondria okadai
157 – Please use Italics to refer to any species name. In this case Psammaplin aplysilla
158 – Please indicate here or in the table the other species.
186 – increase in survival rate of how much %? Please complete.
203 – Please correct as suggested: “For these reasons, THERE is a strong effort….”
209 - Please use Italics to refer to any species name - Aplidium albicans
212 - Please use Italics to refer to any species name - Trididemnum solidum
215/216 – Please also consider other production systems like on site aquaculture cultivation of these species or even biotechnological production
221 – MM – Multiple Myeloma? If yes use full spelling here and abbreviation in brackets as it is first time you mention it.
233 - Please use Italics to refer to any species name - Ecteinascidia turbinata
315 – Please further expand or clarify this information as it is mentioned several limitations in the paragraph above, but none is clearly contextualized or expanded. Furthermore, the authors do not provide any evidence of epigenetic mechanisms previous data for this symbiosis and the sentence feels out of context here.
351-425 – To all these molecules and drugs the authors should, as done for all other molecules, refer in the text part their marine origin and something about the discovery process as this is a crucial component of this review. Mention in the table is as it is for the other sponge and tunicate molecules too.
372 – Please provide the % of longer survival rate of the trial mentioned
483 – “aspected” should be “expected”
491 – there is a repetition of “we have” in the beginning of this sentence as well as an extra “and” in the same line (“… also predicted the AND toxic effects…”)
550 – Adaptative is misspelled
553 – The authors claim that “the data obtained is nit highly significative” and then suggest further investigations. I would recommend either removing this sentence as a whole or, if it is staying then they must expand on the affirmation made and what is suggested to and how investigated further.
598-600 – This initial sentence may feel misleading as it implies that cancer drug discovery has turned empirical in the more recent years. Instead, I believe the authors wish to emphasize that in recent years drug discovery has been growing in complexity and therefore more and more we are witnessing a combination of empirical, bioinformatics and lab based/clinical trial research going hand in hand to gather a wealth of information for each tested drug candidate. Therefore, if this is true I recommend rewriting this paragraph.
Figure 1 – It would be better to have an image of each marine species with its specific compounds. Alternatively a summary table of all compounds mentions, with all marine origins, marine species, date of discover and cancer application (specific tumors) would be very positive for the readers and a good tool for citation and reference to experts in these fields.
Table 1 – Incorrect spelling of Halichondria okadai. The name of the sponge species where Panobinostat was isolated is different in Table one and that mentioned in line 157. Coeherence shall be maintained or mention all the sponges where it has been identified so far.
Table S1 – Could be useful to have in this table too, the marine origin and original species of each compound to have a full overview for example.
I send the pdf of the manuscript with underlined in blue the relevant parts I comment here for easier detection

Reviewer 2 Report
The manuscript by Santaniello et al. provides an overview of the current state of clinically approved anticancer drugs from marine organisms followed by several predictive bioinformatic approaches to assess their activities and actions. While the title emphasizes the bioinformatic analyses, more of the manuscript is devoted to reviewing the state of approved marine drugs; the title needs to be rewritten to reflect that.
Major
The manuscript is neither review nor research article as it has aspects of both. It appears to be intended as a review, but if that’s the case, the data analysis needs to be removed. If the data analysis is retained, it should be submitted as a research article and the review material considerably condensed for use as the introduction.
The review of marine approved cancer drugs appears to derive much of its information from marinepharmacology.org and also repeats material found in another recent review (DOI 10.3390/md20100636). This is a good reason to revise the manuscript as a research article with only a brief review of the clinical material for the introduction. If this review material is retained, some effort needs to be made to distinguish this review from the others.
The in silico results are problematic in the sense that the authors have submitted clinically approved drugs to artificial intelligence pipelines. Those pipelines were developed based on training sets that included, for example, clinically approved drugs. It’s not surprising that the three compounds analyzed had high scores, since they were likely part of the training set. These bioinformatic platforms are meant to direct a path of research, not validate already validated hits. In the context of directing research, bioinformatic predictions require validation – they are not publishable results on their own. To be used in this manuscript, the predictions need validation.
Plitidepsin is not approved by the FDA. Either that metabolite should be removed from the manuscript or discussion of FDA approval should be removed throughout the manuscript.
Minor
The manuscript has many typographical and grammatical errors. A careful reading is required to select correct words and spelling (e.g., ‘climatic’ on line 79; ‘promotive’ on line 160; ‘repropose’ on line 205; ‘eucaryotic’ on line 217; ‘trascurable’ on line 440; ‘citotoxicity’ line 438, etc.). Organism binomials (and Candidatus) should be italicized and compound names should not be capitalized. Many citations are incorrect (citation 13 on line 163 is incorrect, as is 65 on line 235, 29 on line 239, etc.), and others missing (i.e., isolation of plitidepsin and trabectedin both need a citation, the number of marine molluscs needs a citation, etc).
Hyperbole needs to be removed. Bioactive compounds from marine sources are not ‘exponentially intensifying’ (line 13); nobody expects a ‘miraculous drug’ (line 21), the sea is not an inexhaustible source of organisms (line 38) and the number of marine drugs is quite limited, it’s not a plethora (line 54). Please make a concerted effort to review your work and choose accurate descriptive words.
The passage beginning on line 622 and ending on line 626 should be removed – it’s irrelevant to the discussion. Same with lines 311-317.
Line 615 – there’s much more to ecological relationships than secondary metabolites. Please re-phrase that.
Line 47 – your discussion of the kingdoms of life should be avoided as there is little current consensus of how many kingdoms there are, and even if e.g. viuses are organisms.
Line 195 – the distribution of tunicates among depths does not ‘characterize them for their variable tolerance to the surrounding climatic conditions’. Please re-phrase that.
Line 204 – the discussion of characterizing new species and new molecules in the ‘shortest possible time’ does not make sense.
Line 302 – the number of marine molluscs does not vary, the accounting of them does.
The manuscript invokes epigenetics repeatedly, including an entire section (2) that seemingly has nothing to do with either clinically used marine metabolites nor bioinformatic analyses. Most of the discussion of epigenetics could be deleted.
Table 1, eribulin has a duplicate entry in the MOA column.
The images from the GO predictive analysis are highly pixilated – they should be replace by high-quality images.
The citations are a mess. Some authors are missing, some journal names missing, some journal names are abbreviated, some are not, sometimes sentences are all caps, sometimes not. Please follow the Marine Drugs format for citations.
